# Post-Treatment of Reclaimed Municipal Wastewater through Unsaturated and Saturated Porous Media in a Large-Scale Experimental Model

Reza Tahmasbi [1], Majid Kholghi [2], Mohsen Najarchi [1,*], Abdolmajeed Liaghat [2] and Reza Mastouri [1]

[1] Department of Civil Engineering, Faculty of Engineering, Islamic Azad University of Arak Branch, Arak 38135-567, Iran; reza.tahmasbi1400@gmail.com (R.T.); r-mastouri@iau-arak.ac.ir (R.M.)
[2] Department of Irrigation and Reclamation Engineering, Agricultural and Natural Resources Campus, University of Tehran, Karaj 31584-4314, Iran; kholghi@ut.ac.ir (M.K.); aliaghat@ut.ac.ir (A.L.)
[*] Correspondence: m-najarchi@iau-arak.ac.ir

**Abstract:** In recent decades, groundwater overexploitation has caused an important aquifer level decline in arid zones each year. In addition to this issue, large volumes of effluent are produced each year in metropolitan areas of these regions. In this situation, an aquifer storage and recovery system (ASR) using the reclaimed domestic wastewater can be a local solution to these two challenges. In this research, a post-treatment of reclaimed municipal wastewater has been investigated through unsaturated–saturated porous media. A large-scale, L-shaped experimental model was set up near the second-stage wastewater treatment plant (WWTP) in the west of greater Tehran. The water, soil, and treated wastewater of the experimental model were supplied from the aquifer, site, and WWTP, respectively. The 13 physicochemical parameters, temperature and fecal coliform were analyzed every 10 days in seven points for a period of four months (two active periods of 40 days with a 12-h on–off rate (wet cycles) and a rest period of 40 days (dry cycle) between the two wet cycles). The results showed that the effects of the saturated zone were twice as great as those of the unsaturated zone and two-thirds of the total treatment efficiency. Furthermore, a discontinuous wet–dry–wet cycle had a significant effect on effluent treatment efficiency and contaminants' reduction. In conclusion, an aquifer storage and recovery system using treated wastewater through the unsaturated–saturated zones is a sustainable water resource that can be used for agriculture, environmental and non-potable water demands.

**Keywords:** treated wastewater; effluent; soil aquifer treatment; aquifer storage recovery



## 1. Introduction

The aquifer overexploitation causes several environmental effects, such as a decline in groundwater level and land subsidence [1]. Artificial groundwater recharge is classified as direct surface, direct subsurface, the combination of surface–subsurface, and indirect techniques [2,3].

In Orange County, California, advanced treated wastewater was used for the aquifer recharge [4]. Tyagi and Sharma, 2014, concluded the prerequisites for successful artificial recharge of groundwater were as follows: (1) suitable aquifer, (2) available water sources, and (3) good quality of recharge water [5]. India is one of the pioneer countries in artificially recharging aquifers (4.0 km$^3$/year) to help groundwater replenishment [6]. The human activities leading to the enhanced aquifer recharge can be categorized as (1) unintentional recharge, (2) unmanaged recharge, and (3) managed recharge [7]. Reclaimed wastewater is a new, valuable water source and contains sufficient nutrients for agricultural activities and desertification control [8].

Managed aquifer recharge (MAR) plans to use surface basins or injection wells for the intentional recharge of rainwater and reclaimed water to groundwater for future recovery

or other environmental uses [7]. The use of the surface basin method is a practical choice to recharge wells based on a pilot test [9]. The MAR projects are growing worldwide due to their need for fewer storage sites, water shortages, high evaporation rate, and low costs [10]. Fichtner et al. (2019) simulated several MAR scenarios, such as a 1D-lab column, a 3D-lab tank, and a 3D-field unit, to understand the impact of the experimental models on the evaluation of activities and to find a suitable experimental setup to clarify these processes, and they finally recommended 3D experiments due to their more realistic display of flow processes [11]. Bekele et al. (2018) showed that MAR is a low-cost alternative to the other wastewater treatment methods [12]. Samanta et al. (2020) used Mod flow software to simulate the MAR method and showed a 30% reduction in hydraulic conductivity in the infiltration basin bed in the winter compared to the summer [13]. In recent years, many MAR methods have been developed for artificial groundwater recharge, such as soil aquifer treatment (SAT), aquifer storage and recovery (ASR), and aquifer storage, transfer, and recovery (ASTR).

In the MAR system using a surface method, the consideration of wet and dry periods leads to improved efficiency due to the aerobic conditions provided in the porous media [14]. Regarding the dry and wet periods, artificial recharge basins are classified as (1) continuous flow, (2) dry and wet cycle, and (3) filling and emptying basins [15,16]. Hammer (2016) used the filling and emptying basins in Florida, US, in which the basins were filled to a depth of from 1 to 3 m and then the water was allowed to completely infiltrate the ground (wet cycles) [15]. Finally, the ponds were left to dry out and higher efficiencies were reported for the water infiltration rate in the aquifer using this method. Wintgens et al. (2009) concluded that, in the MAR methods, economic aspects should be considered along with challenges such as resolving water shortage, drought management, and the environmental goals [17]. Bekele et al. (2011) studied the infiltration of secondary-stage treated wastewater through a vadose zone of 9 m thickness for 39 months using an MAR method to define the potential improvements in recycled water quality. The average concentrations of several constituents, including phosphorous, fluoride, iron, and total organic carbon, decreased by 30%, 66%, 62%, and 51%, respectively. Effluent was entered at a flow rate of 17.5 L/min, with a residence time of four days in the vadose zone and about two days in the saturated zone [18]. Du et al. (2014) studied a series of soil column experiments to show the clogging process that was caused by suspended solids, and categorized the process as surface clogging, internal clogging, and mixed clogging [19]. Thangarajan (2007) recommended a relative reduction in suspended solids to about 10 mg/L for groundwater recharge projects [20]. Alam et al. (2021) studied 1127 MAR sites and concluded that the MAR has the best efficiency in sandy clay loam soils [21].

The use of soil and aquifer as natural treatment systems is called soil aquifer treatment (SAT) and is an advanced treatment system using a simple, inexpensive technology [14]. Bacterial mobility plays a vital role in the transfer of contaminants and bioremediation in the unsaturated zones [22]. The unsaturated zone can remove all suspended solids, biodegradable materials, and microorganisms [14]. Using a large-scale physical model, Schaffer et al. (2015) studied the effects of a compost layer on the reduction in 28 micro-contaminants under SAT systems, and their results showed the highest reduction in organic cations [23]. El Arabi (2012) indicated that SAT could be used as a proper alternative to the direct use of wastewater due to the increasing water demand [24]. Amy and Drewes (2007) tracked the removal of effluent organic matter using the SAT system and showed that the total removal rate significantly improved [25].

Aquifer Storage and Recovery (ASR) systems or well injection techniques are used in urban areas where the available land is not sufficient to use surface basin systems or when confined aquifers are very deep [6]. The Aquifer Storage, Transfer, and Recovery (ASTR) method brings the wastewater to the aquifer for storage, transferring it to the groundwater, and recovering using other supply wells. This method leads to an improvement in the treated wastewater quality due to the wastewater's longer residence time in the aquifer and the completion of some chemical, biological, and physical processes [26]. Before

reclamation and reuse, the injected effluent needs a residence time of a few months, and sometimes several years, depending on the effluent quality and aquifer properties [27].

Reese (2002) studied the ASR system on 23 sites in Florida, US, and addressed the vital hydrogeological factors in the design and management of an ASR system [28]. Dillon et al. (2006) studied two ASR systems in Australia, one recharged through injection wells and the other recharged using the infiltration basin method, and indicated the limitations and requirements of both approaches [29]. Castro (1996) suggested that the ASR projects are a cost-effective and efficient solution to increase the aquifer capacity and the removal of contaminations from the effluent [30]. Asano (2006) showed that the four qualitative factors, including microbiological quality, total dissolved solids (TDS), dangerous heavy metals, and concentration potential of harmful organic matters, should be considered when using the effluent for aquifer recharge [31]. Many organic micropollutants ($OMP_S$) can resist in the WWTP [23].

Sheng (2005) injected $74.7 \times 10^6$ m$^3$ reclaimed water, using ten injection wells, into the aquifer of Hueco Bolson for 18 years and found that a distance of approximately 782 m should be considered between recovery and injection wells to provide enough residence time for the recharged water in the saturated zone [32]. Rinck-Pfeiffer et al. (2013) studied various reclaimed water effects on hydraulic conductivity (K) using three laboratory columns, and the results showed a decrease in hydraulic conductivity in all columns due to the clogging problems of the injection wells [33]. Stuyfzand et al. (2005) studied the ASR system using injection wells at the Herten site, Netherlands, and they showed that ASR was hydrologically feasible on that site because water could be injected and recovered at a high rate without clogging problems [34]. Surface clogging, and the subsequent reduction in infiltration rate, are major problems in the artificial recharge of aquifers through the infiltration basins, so the hydraulic loading rate should be controlled in proportion to the infiltration rate in the basin [35]. Voudouris (2011) studied borehole clogging in an aquifer recharge project using treated wastewater at the Mesaoria, Cyprus, and pointed out that clogging occurs due to (1) the high concentration of recharge water, (2) bacterial growth caused by high amounts of Assimilable Organic Carbon (AOC) in the recharge water, (3) chemical reactions, and (4) suspended solids (SS) [27].

Pavelic et al. (2007) [36] studied an ASR site at Bolivar in South Australia for more than four years. They showed that low clogging rates could be achieved for turbidity < 3 NTU, pH < 7.2, and total nitrogen < 10 mg/L. Injectant and groundwater at the 4 m observation well were sampled at 7–14-day intervals for a detailed suite of physicochemical parameters. The injection/recovery wells were backwashed by pumping the rate of injection 2–3 times for one hour [36]. Händel et al. (2014) [37] showed that the ASR systems using the surface basins or large-diameter injection wells are usually costly and developed a cost-effective ASR method. They suggested groundwater recharge using gravity in small-diameter wells, which were drilled with the direct-push technique. They showed this approach possesses a distinct advantage over recharge by surface basins [37].

The present study aims to further investigate the ASTR system and utilize its comparative advantages in comparison to the other MAR methods for the post-treatment of secondary-stage treated wastewater through the unsaturated and saturated zone at a large-scale and pre-pilot experimental model. The experimental model was designed and implemented based on the ASTR system and fifteen physiochemical parameters, including T, pH, $BOD_5$, COD, DO, $CL^-$, $NO_3^-$, $PO_4^{3-}$, EC, TH, $Ca^{2+}$, $Mg^{2+}$, $Na^+$, SAR, and fecal coliform, were sampled and analyzed over four months. The pre-pilot experimental model tests are beneficial, which is usually representative of the site situation. The large-scale experimental model used in this research is much better than laboratory models (lab columns) and its performance is lower than the pilot studies and can be developed to pilot condition. Better results would have been obtained with a longer study period and higher soil column on the experimental model.

## 2. Materials and Methods

In this study, a large-scale, L-shaped experimental model was installed immediately after the secondary treatment stage of the Qods City wastewater treatment plant (WWTP) in the west of Tehran. Water, soil, and effluent (treated wastewater resource of the experimental model) were supplied from the aquifer, site, and WWTP, respectively. The main steps of this research are shown in Figure 1.

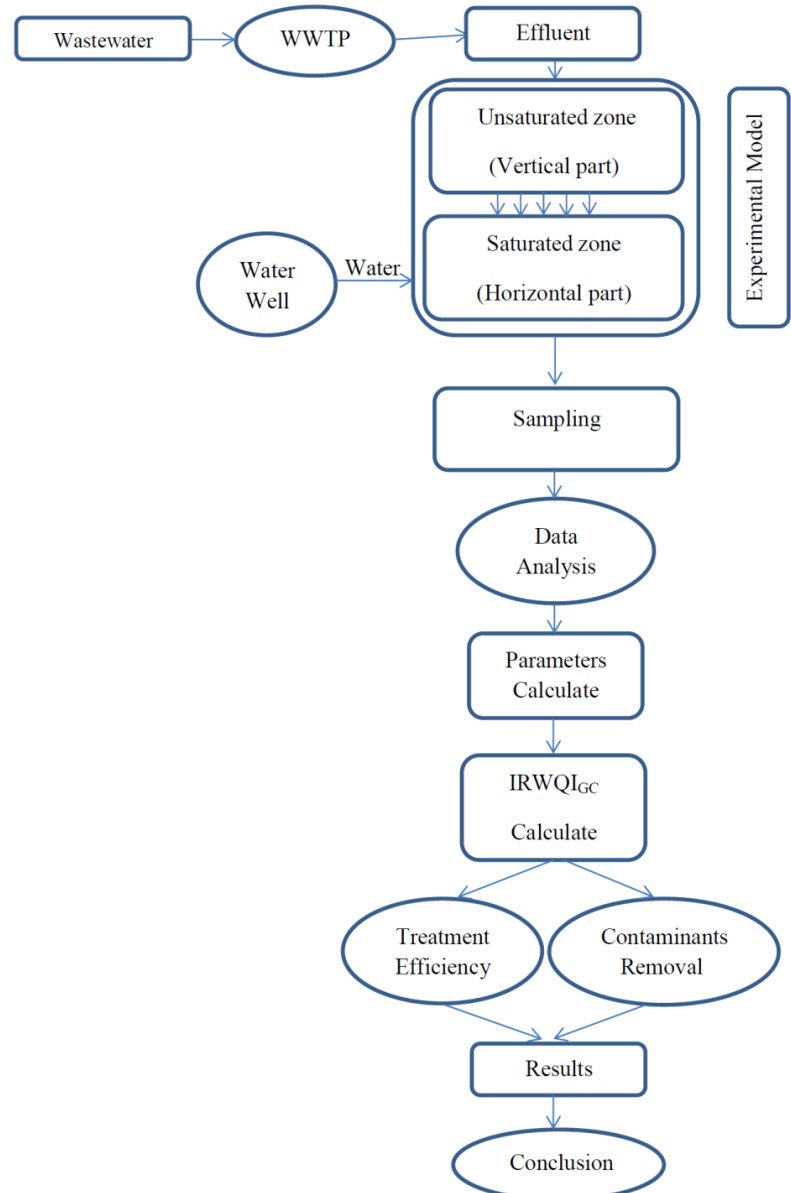

**Figure 1.** Study Diagram.

### 2.1. Facilities

The municipal treated wastewater (secondary stage) was supplied from the Qods City WWTP in the Shahryar plain, located in the west of Tehran, Iran. The treated wastewater at the WWTP outlet was pumped to the experimental model and was passed through the filter layer, and unsaturated and saturated media, in that order. The soil and water data of the experimental model were obtained from Shahryar's plain. The soil samples, grain size, and physicochemical properties were analyzed and are shown in Tables 1 and 2. The aquifer water and treated wastewater qualitative properties are shown in Tables 3 and 4, respectively.

**Table 1.** Sieve analysis of soils (ASTM D422).

| Sample No. | Unified Soil Classification | Description | Liquid Limit (%) | Plasticity Index (%) |
|---|---|---|---|---|
| S1 | ML | Silt with Sand Cc = 1.242 and Cu > 10 | 23 | NP |
| S2 | ML | Silt with Sand Cc = 1.120 and Cu > 10 | 24 | NP |

**Table 2.** Results of the chemical tests on the soil samples.

| Parameter | PH | Total Hardness. (CaCo$_3$) (ppm) | EC (micromhos/cm) | Total N (T.K.N) (ppm) | Na (Saturated Extract) (ppm) | Total Phosphorus (ppm) | Mgo (%) | Cao (%) | Chloride (%) |
|---|---|---|---|---|---|---|---|---|---|
| **Amount** | 7.15 | 195 | 222 | 0.1 | 66.7 | 0.046 | 2.1 | 7.99 | 0.009 |

**Table 3.** Quality of aquifer water used in the experimental model.

| Parameters | PH | Turbidity (NTU) | TH (mg/L) | Alkalinity (mg/L) | EC (μs/cm) | TDS (mg/L) | CL (mg/L) | SO$_4$ (mg/L) | NO$_3$ (mg/L) | NO$_2$ (mg/L) | PO$_4$ (mg/L) | Ca (mg/L) | Mg (mg/L) | Na (mg/L) | K (mg/L) | NH$_3$ (mg/L) |
|---|---|---|---|---|---|---|---|---|---|---|---|---|---|---|---|---|
| *Amount* | 7.7–7.8 | 0.2–0.5 | 326–392 | 104–110 | 922–1100 | 525–605 | 156–178 | 85–105 | 26–43 | 0.003–0.007 | 0.37–0.38 | 102–121 | 16.7–18.2 | 55.1–57.5 | 0.5–0.6 | 0.05–0.07 |

**Table 4.** Treated wastewater (effluent) quality/WWTP outlet/(January–June) 2018.

| Parameters | BOD$_5$ (mg/L) | COD (mg/L) | TSS (mg/L) | TDS (mg/L) | NO$_3$ (mg/L) | TP (mg/L) as P | TN (mg/L) as N | Total Coliform | PH | T (°C) |
|---|---|---|---|---|---|---|---|---|---|---|
| *Range* | 4.3–7.10 | 18.0–28.0 | 5.0–9.0 | 827–908 | 4.4–24.7 | 1.15–3.13 | 10.0–15.6 | 110–1600 | 7.4–8.2 | 12.7–24.7 |

### 2.2. Experimental Model

### 2.2.1. Model Installation Site and Wastewater Treatment Plant (WWTP)

In this research, the experimental model was installed immediately after the outlet of the Qods City WWTP and recharged with its effluent. The WWTP with a 12-ha area is located about 20 km from Tehran, at $51°07'$ east longitude and $35°43'$ north latitude. The treatment method is based on the BIOLACK-activated sludge base process. This WWTP, with a total anticipated capacity of 114,600 $m^3$/day, was designed in three modules; the first module was implemented with a capacity of 28,650 $m^3$/day. At the time of the study, the other two modules had not been implemented yet due to insufficient input flow. The WWTP contains the following parts: entrance gate, primary screening, grit unit, channels, distribution chamber, pump station, BIO-Pond, chlorination contact tank, chlorination building, supernatant, sludge thickener, and dewatering building.

### 2.2.2. Experimental Model Design

The experimental model was installed immediately after the secondary-stage treated wastewater plant in the west of Tehran. The unsaturated zone (vertical part of the model) was simulated using a polyethylene pipe with a height of 3.0 m and a diameter of 1200 mm. The saturated zone (horizontal part of the model) was simulated using a UPVC pipe with a length of 20 m and a diameter of 315 mm. Additionally, a 30-cm thickness filter layer made of igneous rock fragments was used on top of the unsaturated zone. The water, soil, and treated wastewater resources of the experimental model were supplied from the aquifer, site, and WWTP, respectively. The conceptual model of this setup is illustrated in Figure 2.

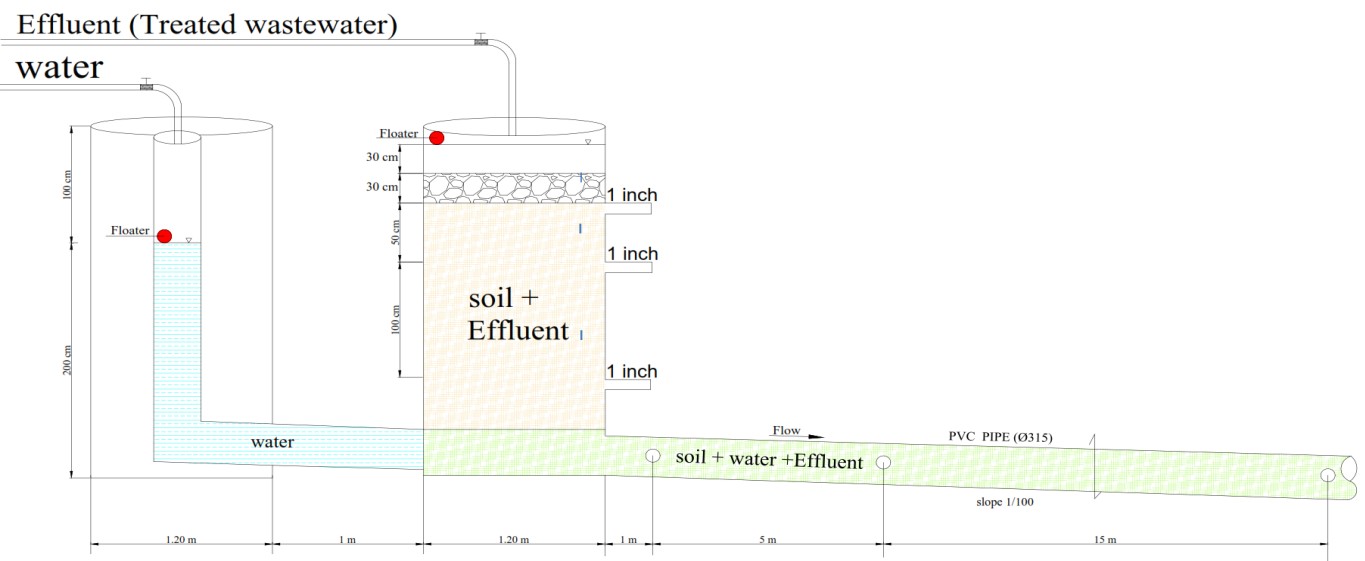

**Figure 2.** Schematics of the experimental model.

### 2.3. The Hydraulic Conditions

This study was conducted over four months, and comprised two 40-day recharge (wet) cycles and one 40-day rest (dry) cycle between the wet cycles. An on/off system was used on the flow during each wet cycle (12-h on and 12-h off). Two kinds of 30-cm filter layers were installed on the top of the soil column: the first one was the igneous rock fragments with $d_{50}$ of 20 mm, and the second one was the marble stone fragments with $d_{50}$ of 10 mm. In this study, the discharge rate of outlet wastewater was 0.12 L/min, and the hydraulic conductivity was 0.005 cm/s. A 30-cm constant wastewater head was considered on the soil column. Three manometers were used to measure water head in the vertical chamber in 0.3, 0.8, and 1.8 m and three manometers in the horizontal pipe in 1, 5, and 20 m.

*2.4. Collection and Analysis of Water Samples*

Fifteen physiochemical parameters (shown in Tables 5 and 6), including T, pH, $BOD_5$, COD, DO, $Cl^-$, $NO_3^-$, $PO_4^{3-}$, EC, TH, $Ca^{2+} + Mg^{2+}$, $Na^+$, SAR, and fecal coliform, were sampled and analyzed every ten days at seven points of the experimental model. These seven points included the input of the effluent, area immediately after filter layer, unsaturated zone (0.5 and 1.5 m), and saturated zone (1, 5, and 20 m). For each part, a 25-mm polyethylene pipe and valve were installed for sampling and flow control. Furthermore, an end cap was used to close the end of the horizontal part. All fifteen parameters for all the specimens sampled over the four months at the 7 points of the experimental model were measured and analyzed in a laboratory setting. Then, the results were compared to the Iran Water Quality Index for groundwater conventional parameters ($IRWQI_{GC}$).

**Table 5.** The weight of each parameter and units (IRWQI).

| Parameters | Weight of Each Parameter |
|---|---|
| T (°C) | - |
| PH (Standard unit) | 0.074 |
| $BOD_5$ (mg/L) | 0.09 |
| COD (mg/L) | 0.08 |
| DO (Saturation %) | 0.067 |
| SAR | 0.089 |
| $Cl^-$ (PPM) | - |
| $NO_3^-$ (mg/L) | 0.151 |
| $PO_4^{2-}$ (mg/L) | 0.085 |
| EC(µ Siemens/cm) | 0.129 |
| Total Hardness (mg/L $CaCo_3$) | 0.103 |
| Fecal Coliform (MPN/100 mL) | 0.134 |

**Table 6.** The guide table used to determine the descriptive equivalent of a calculated index.

| The Amount of Index ($IRWQI_{GC}$) | The Descriptive Equivalent of Water Quality |
|---|---|
| <15 | Very Poor |
| 15–29.9 | Poor |
| 30–44.9 | Fairly Poor |
| 45–55 | Medium |
| 55.1–70 | Fairly Good |
| 70.1–85 | Good |
| <85 | Excellent |

The Iran Environmental Protection Agency (IR EPA) guideline is used in this study to calculate $IRWQI_{GC}$. This index is obtained from the combination of the British Colombia Water Quality Index (BCWQI) and the National Sanitation Foundation Water Quality Index (NSFWQI). Groundwater quality is based on the influence of pollutants released to the environment from human activities [38]. WQI was developed by selecting water quality variables such as pH, dissolved oxygen, specific conductance, alkalinity, chloride, coliforms, etc. [5]. WQI is an effective tool to provide feedback on water quality to policymakers and environmentalists [39].

## 3. Results and Discussion

$IRWQI_{GC}$ is an appropriate method to determine the qualitative conditions of groundwater in which the data related to the several water quality parameters were included in a mathematical formula that showed a quantitative representation of the water quality status. Using this method, a number is used to show the water quality status, which is categorized in a relative scale that indicates water quality from very poor to excellent [40]. Descriptive equivalents of the calculated indices are determined according to Tables 5 and 6. The analysis of 1050 datasets (five tests in each wet cycle in 10-day intervals, resulting in 70 treated wastewater samples from seven marked points, and also laboratory analysis of fifteen quality parameters), then $IRWQI_{GC}$ index was calculated by Equation (1) based on the IRWQI guideline

$$IRWQIGC = \left[ \prod_{i=1}^{n} I_i^{W_i} \right]^{\frac{1}{\gamma_i}}$$
$$\gamma_i = \sum_{i=1}^{n} W_i$$

(1)

where: $W_i$—weight of parameter i, n—number of parameters, $\gamma_i$—index value for parameter i.

For all the parameters, the removal rates were determined, and the results showed a considerable decrease in most parameters. The amount of $Na^+$, $(Ca^{2+} + Mg^{2+})$, SAR, total hardness (TH), EC, $PO_4^{3-}$, $NO_3^-$, COD, $BOD_5$ and fecal coliform decreased by 20.81%, 12.48%, 15.03%, 12.48%, 9.07%, 65.42%, 26.44%, 23.73%, 23.27% and 73.20%, respectively (Figures 3–12).

As shown in Figure 12, the linearization of the coliform variation curve in one of the experiments (Test 5) was due to the high concentration of the effluent inflow to the model and the system's inability to reduce the contaminations.

According to the IRWQI guidelines, considering a WQI of 85 for excellent quality (see Tables 5 and 6), the results indicate the incremental trend of the effluent treatment efficiency in both zones (Tables 7–9 and Figure 13). The efficiency in the unsaturated zone was about 1.51–7.48%, with an average of 4.53%, and in the saturated zone efficiency was about 1.89–20.21%, with an average of 7.19%. The total efficiency of the effluent under the unsaturated and saturated zones and the effects of the filter layer were between 5.25% and 23.08%, with an average of 12.72%. It should be noted that the effluent treatment efficiency through the filter layer was very small, at about 0.98% (Tables 8 and 9).

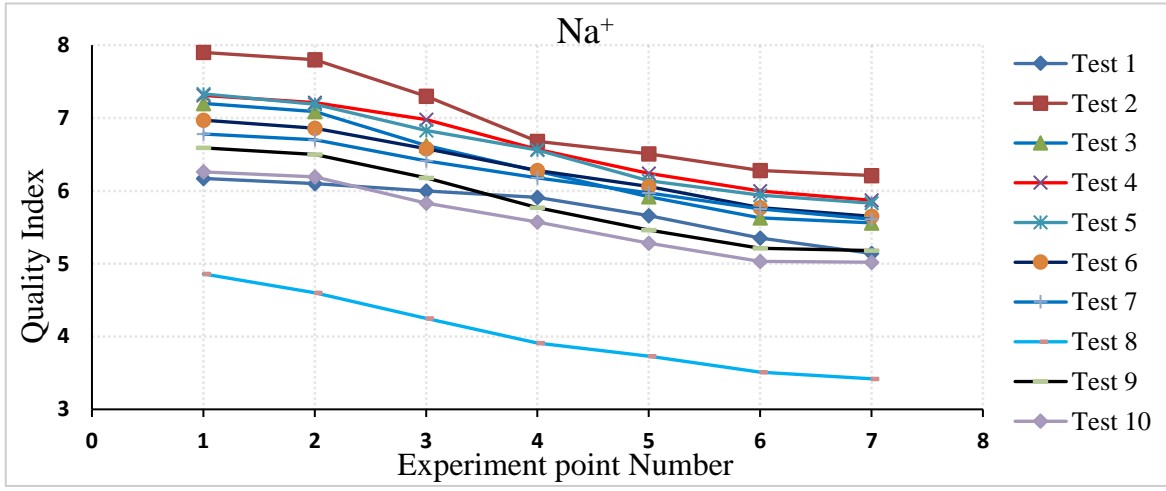

**Figure 3.** The $Na^+$ changes over time at various points.

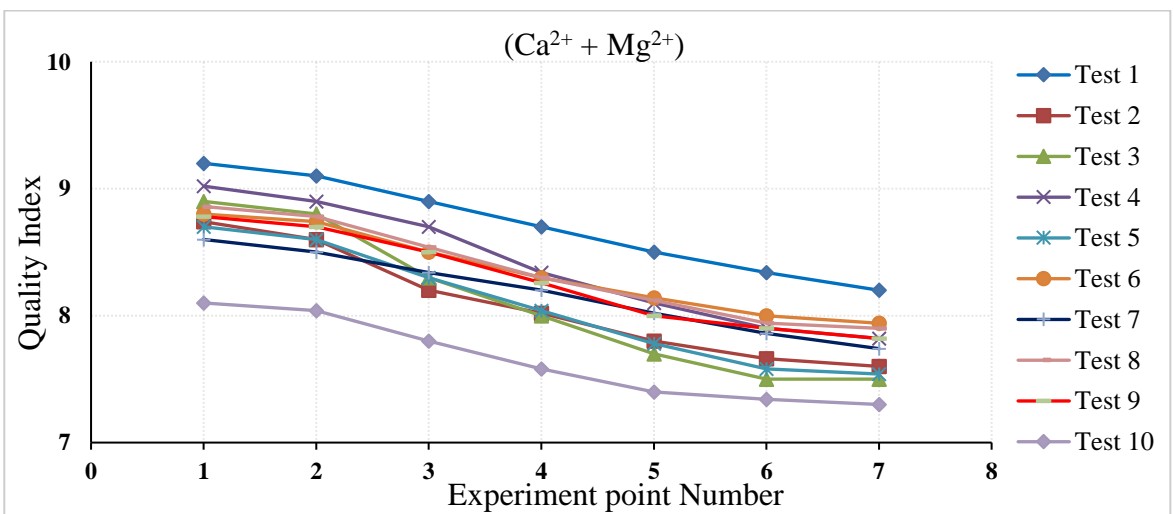

**Figure 4.** The $Ca^{2+} + Mg^{2+}$ changes over time at various points.

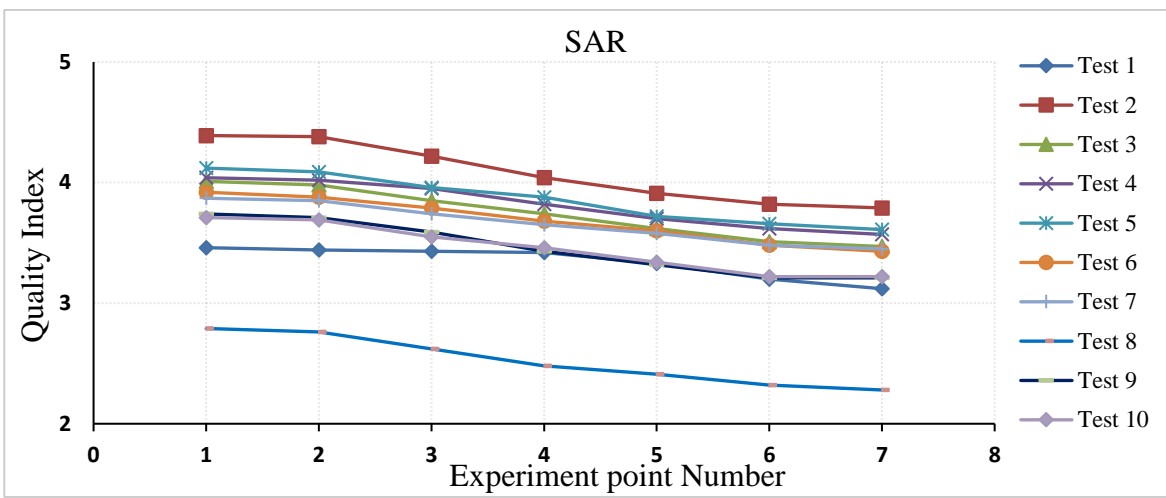

**Figure 5.** SAR Index changes over time at various points.

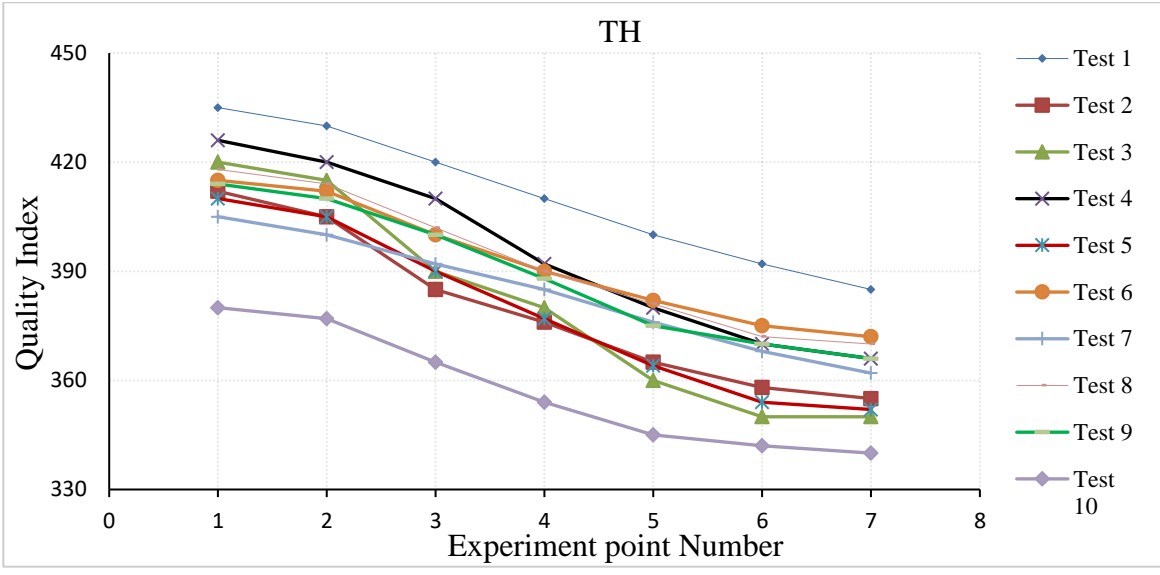

**Figure 6.** Total Hardness Index changes over time.

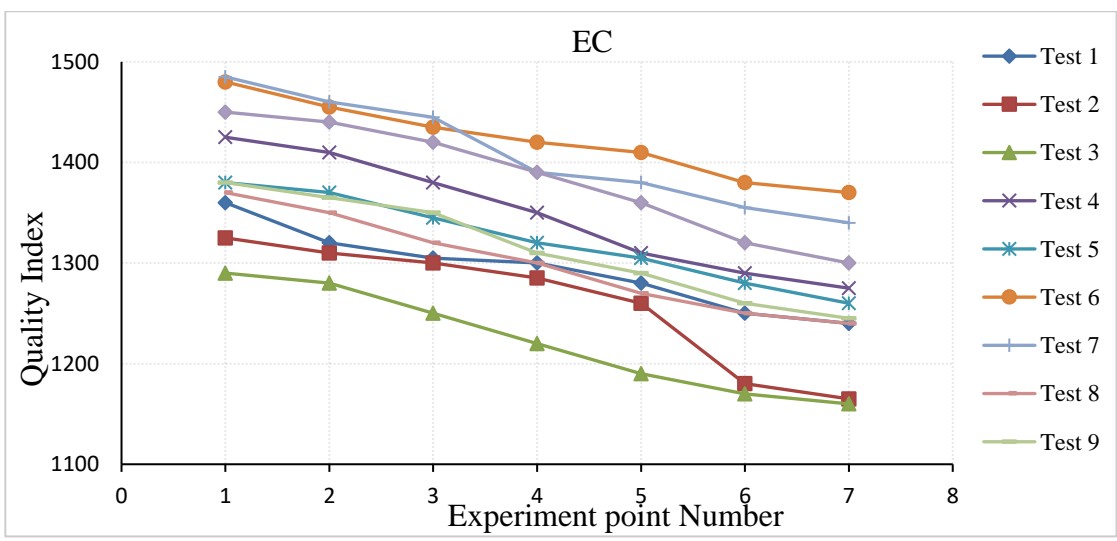

**Figure 7.** EC Index changes over time at various points.

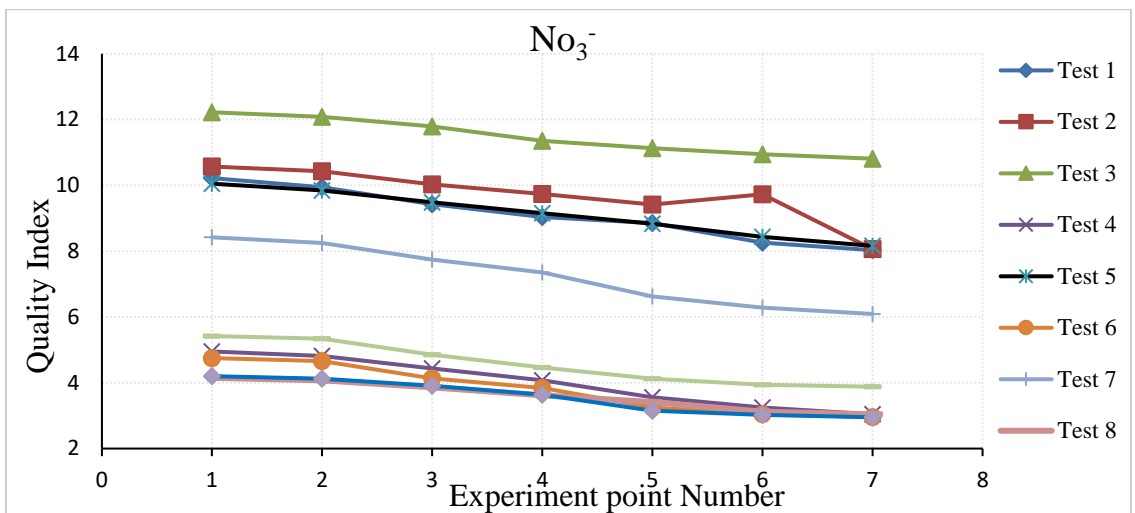

**Figure 8.** $NO_3^-$ Index changes over time at various points.

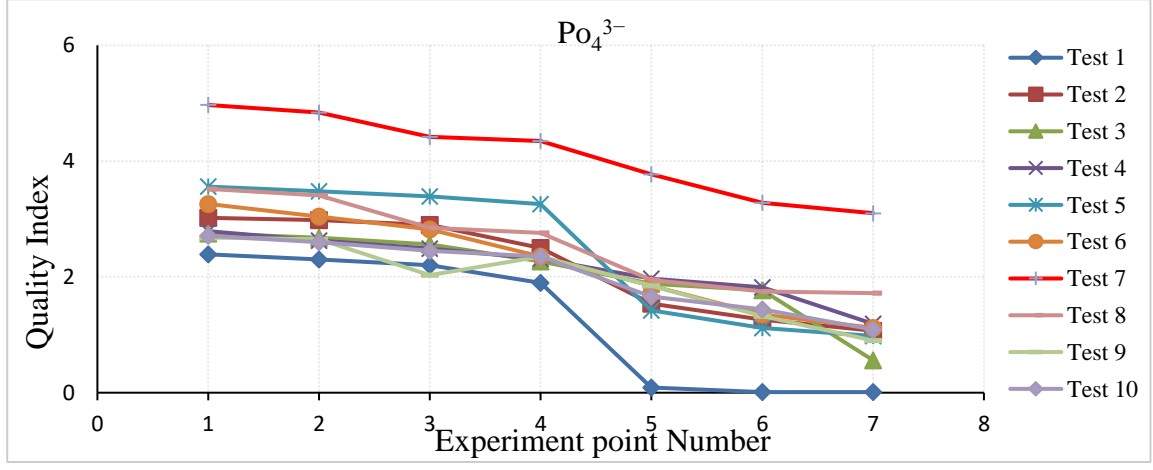

**Figure 9.** $PO_4^{3-}$ Index changes over time at various points.

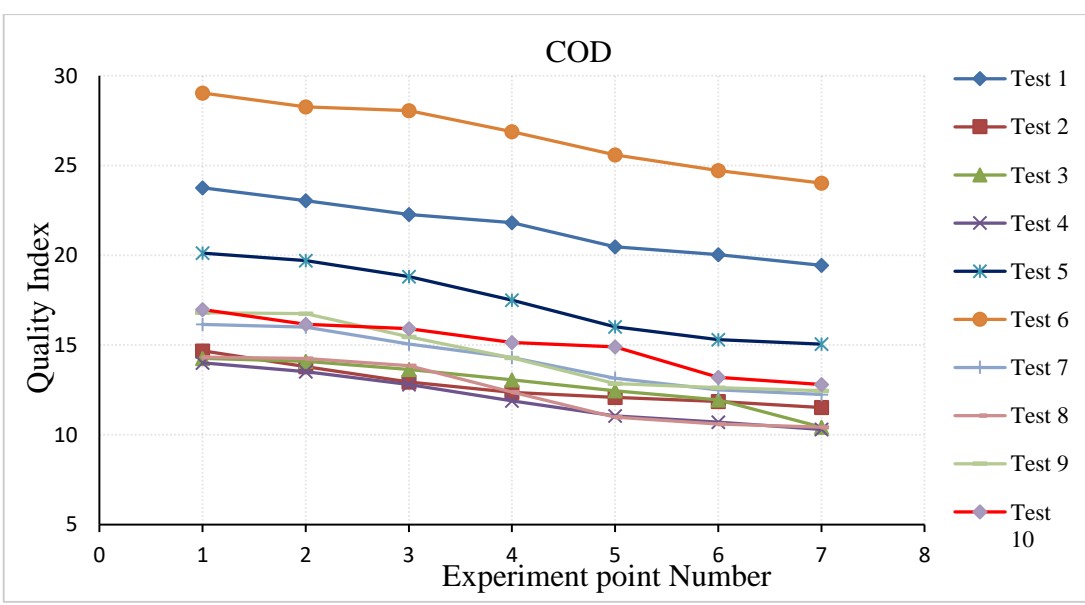

**Figure 10.** COD Index changes over time at various points.

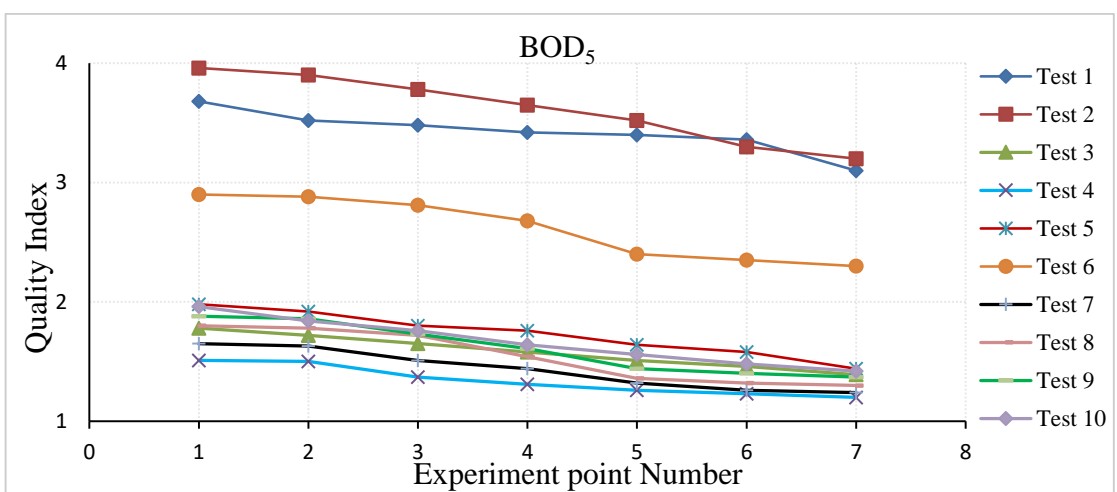

**Figure 11.** BOD$_5$ Index changes over time at various points.

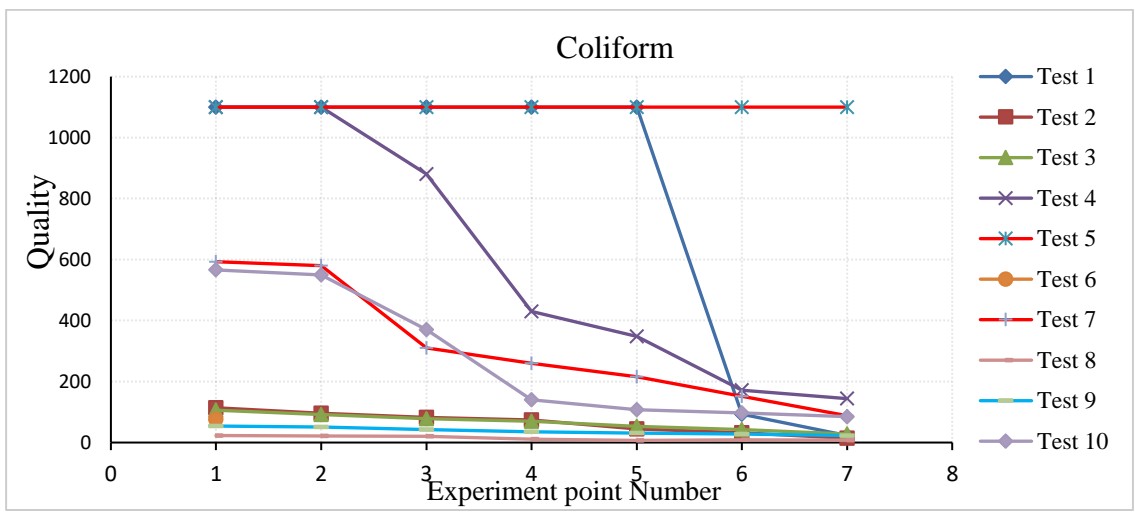

**Figure 12.** Coliform changes over time at various points.

**Table 7.** Treated Wastewater Quality Indices (IRWQI) at marked points of the experimental model in the ten experiments.

| Descriptions | Points | IRWQI of the Test Samples First Recharge Period (1st Wet Cycle) | | | | | Second Recharge Period (2nd Wet Cycle) | | | | |
| --- | --- | --- | --- | --- | --- | --- | --- | --- | --- | --- | --- |
| | | Test1 | Test2 | Test3 | Test4 | Test5 | Test6 | Test7 | Test8 | Test9 | Test10 |
| Effluent | 1 | 22.85 | 29.1 | 31.99 | 26.93 | 23.59 | 22.94 | 26.08 | 36.39 | 33.08 | 30.98 |
| Filter Effects | 2 | 24.01 | 31.07 | 32.5 | 27.04 | 24.36 | 24.01 | 27.04 | 36.38 | 34.18 | 31.8 |
| Unsaturated Zone Effects in 0.5 m | 3 | 24.61 | 32.79 | 34.23 | 29.69 | 25.71 | 26.82 | 29.64 | 38.93 | 35.94 | 33.52 |
| Unsaturated Zone Effects in 1.5 m | 4 | 25.29 | 34.45 | 35.53 | 33.4 | 26.44 | 29.95 | 31.28 | 41.04 | 37.58 | 35.91 |
| Saturated Zone Effects in 1 m | 5 | 28.38 | 35.95 | 38.18 | 36.07 | 28.85 | 32.85 | 33.48 | 43.21 | 40.02 | 37.87 |
| Saturated Zone Effects in 5 m | 6 | 40.15 | 37.53 | 37.96 | 38.64 | 28.38 | 34.57 | 35.68 | 43.87 | 41.57 | 40.27 |
| Saturated Zone Effects in 20 m | 7 | 42.47 | 39.63 | 40.35 | 39.88 | 28.05 | 35.8 | 36.56 | 44.33 | 43.06 | 41.89 |

**Table 8.** Effluent treatment efficiency (with WQI = 85).

| | Efficiency in First Recharge Period (1st Wet Cycle) | | | | | Efficiency in Second Recharge Period (2nd Wet Cycle) | | | | | Average |
| --- | --- | --- | --- | --- | --- | --- | --- | --- | --- | --- | --- |
| Filter Efficiency | 1.36 | 2.32 | 0.6 | 0.13 | 0.91 | 1.26 | 1.13 | −0.01 * | 1.29 | 0.96 | 0.98 |
| Unsaturated Zone Efficiency | 1.51 | 3.98 | 3.56 | 7.48 | 2.45 | 6.99 | 4.99 | 5.48 | 4 | 4.84 | 4.53 |
| Saturated Zone Efficiency | 20.21 | 6.09 | 5.67 | 7.62 | 1.89 | 6.88 | 6.21 | 3.87 | 6.45 | 7.04 | 7.19 |
| Total Efficiency | 23.08 | 12.39 | 9.84 | 15.24 | 5.25 | 15.13 | 12.33 | 9.34 | 11.74 | 12.84 | 12.7 |

* These data were not used in the analyses.

**Table 9.** The range of the treatment efficiency in each part (%).

| Layers or Zones | Treatment Efficiency (%) | Average Treatment Efficiency (%) |
| --- | --- | --- |
| Filter Effect | 0.13–2.32 | 0.98 |
| Unsaturated Zone Effect | 1.51–7.48 | 4.53 |
| Saturated Zone Effect | 1.89–20.21 | 7.19 |
| Total Layers Effects | 5.25–23.08 | 12.7 |

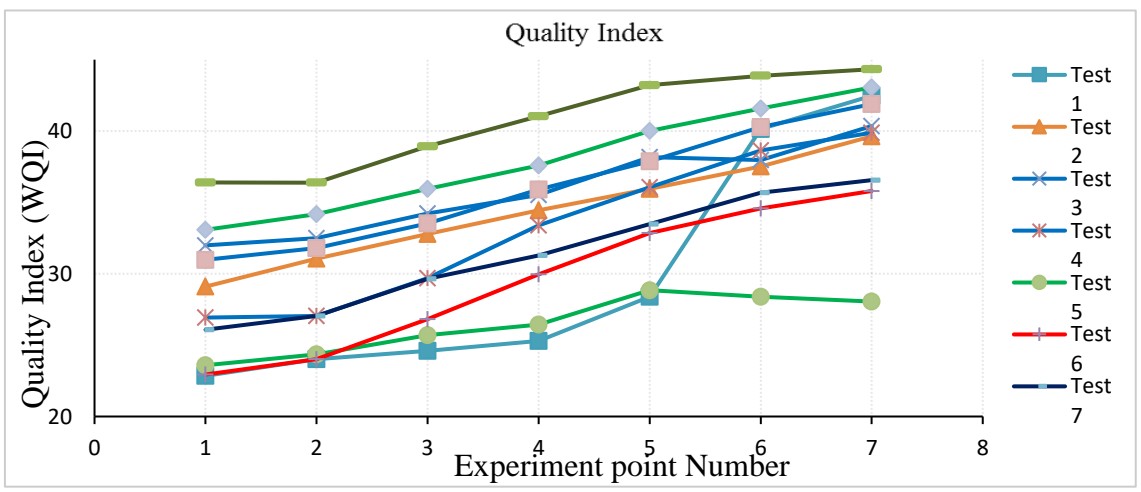

**Figure 13.** Changes of treated wastewater quality index (improvement of WQI).

In this experimental model, the dry cycle and the two wet cycles are considered for the experiments. The results showed that, in the 40-day rest period (dry cycle), the treatment efficiency increased from 5.25% at the previous stage up to 15.13% in the second stage; in other words, there was an increase of about 9.88%. This indicates that we can obtain a higher efficiency if the wet period decreases from 40 days to 30 days.

According to Table 10, the removal efficiencies of vital parameters such as COD, $BOD_5$, SAR, TH, $NO_3^-$, $PO_4^{3-}$, EC and Fecal Coliform were 36.3%, 10.6%, 19.7%, 31.5%, 6.9%, 55.4%, 14.2 % and 48%, respectively.

**Table 10.** The removal efficiency for several vital parameters.

| Parameter Index | COD | $BOD_5$ | SAR | TH | $NO_3^-$ | $PO_4^{3-}$ | EC | Fecal Coliform |
|---|---|---|---|---|---|---|---|---|
| *Range* | 21.1–42.7 | 4.8–17.9 | 15.0–26.9 | 23.3–37.5 | 3.6–10.6 | 40.8–91.8 | 10.3–20.0 | 15.5–88.9 |
| **Average** | 36.3 | 10.6 | 19.7 | 31.5 | 6.9 | 55.4 | 14.2 | 48.0 |

## 4. Conclusions

In recent years, effluent or treated municipal wastewater has been widely considered as a valuable source of water in the aquifer recharge by the ASR system. The effluent requires a post-treatment, especially to achieve environmental goals. The purpose of this research was to investigate the effluent (secondary stage treated wastewater) post-treatment or removal of contaminants through the unsaturated and saturated zones in a large-scale, pre-pilot experimental model. This study evaluated the effects of (i) unsaturated/saturated zones, (ii) wet and dry intervals, and (iii) a filter layer as extra remediation of treated wastewater.

The effects of the saturated zone were twice as great as the unsaturated zone and two-thirds of the total treatment efficiency. This might be due to the excessive length of the horizontal part of the model, the residence time of the effluent, and the dilution compared to the low thickness of the unsaturated zone. The use of the wet and dry cycles had a significant effect on the effluent treatment efficiency and contaminant reduction. In the dry cycle, the system's efficiency increased due to the drying of the unsaturated zone and consequently suitable ventilation (leading to aerobic conditions). On the other hand, when the wet cycle decreases, the post-treatment efficiency tends to increase.

In this research, the filter layer on the top of the unsaturated zone did not show a significant effect on the effluent quality. However, it can help to prevent clogging in unmanaged wastewater inlets due to the inflow of a high load of suspended solids (SS), and reduces the vulnerability of the unsaturated zone. Finally, it was concluded that an ASTR system using treated wastewater or effluent passing through the unsaturated–saturated zones is a sustainable water resource that can be used for agriculture, environmental and non-potable water demands.

**Author Contributions:** Investigation and writing—original draft preparation, R.T.; formal analysis, M.K.; supervision, M.N.; writing—review and editing, A.L.; validation, R.M. All authors have read and agreed to the published version of the manuscript.

**Funding:** This research received no external funding.

**Acknowledgments:** The authors wish to thank the Regional Water Company of Tehran (contract Nr. 3–73-155 with the University of Tehran) for providing some of the financial support, and to thank the Water and Wastewater Company of Tehran province to provide the infrastructure needed to set up the experimental model in the WWTP of Qods City.

**Conflicts of Interest:** The authors declare no conflict of interest.

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
