# Peer review of "Post-Treatment of Reclaimed Municipal Wastewater through Unsaturated and Saturated Porous Media in a Large-Scale Experimental Model"

_water, doi:10.3390/w14071137_

Round 1

Reviewer 1 Report

In this study, the authors have studied a post-treatment of reclaimed municipal wastewater through unsaturated-saturated porous media. For this purpose, they set up a large-scale L-shaped experimental model near the second stage wastewater treatment plant and monitored and analyzed the physicochemical parameters, temperature, and fecal coliform.

The paper is well organized, and the subject is relevant to the scope of the journal. However, it needs to be revised before acceptance.

  • Direction of texts and numbers should be adjusted in Table 3.
  • ‘n’ in Equation (1) should be defined.
  • The authors should clearly mention limitations and weaknesses of the used method and model in the manuscript.
  • Please clarify why the results obtained from test 8 presented in Figures 3, 5 are significantly lower than others.
  • Please clarify why there is no change in the results obtained from tests 5, 8 and 9 over time at various points, presented in Figure 12, unlike the results obtained from other tests.

Author Response

Dear Reviewer

First and foremost, we would like to take this opportunity to appreciate the efforts of the respected reviewers. We have attempted to follow the points carefully and modify the paper accordingly. Through a checklist as below, we have included these modifications.

Best Regards

Mohsen Najarchi

----------------------------------------------------------------------------------------

For Reviewer No. 1:

Some of the comments of the first reviewer that needed to be corrected in the text of the article were made and the necessary edits were made in the text of the article. Regarding the last 2 comments, the necessary explanations are provided as follows:

Comment:

Please clarify why the results obtained from test 8 presented in Figures 3, 5 are significantly lower than others.

Response:

Considering that the effluent of the wastewater treatment plant (with the same conditions and quality) was used as the input of the experimental model and the researchers did not interfere in this data and its fluctuations were related to the operation of the treatment plant and special conditions of treatment processes in the day of measurement. Therefore, the data used in the curves of Figures 3 and 5, including the data related to test 8, have all been extracted from the measurement tables of the desired parameters at the outlet of the wastewater treatment plant and have been used as model input. As can be seen in the graphs, the process of change is almost the same as other tests.

Comment:

Please clarify why there is no change in the results obtained from tests 5, 8 and 9 over time at various points, presented in Figure 12, unlike the results obtained from other tests.

Response:

Regarding test number 5, below figure 12 in the text of the article, the reason for the linearization of the curve related to the above test was explained. Regarding tests 8 and 9, due to the vertical scale of the graph, despite the changes, the relevant curve does not show the mentioned changes clearly. In addition, in tests where the amount of input coliform to the experimental model was low, the model did not perform well in order to improve its quality.

We hope that the above statements have addressed the concerns brought up, and the revised manuscript has improved for a positive recommendation by the Referees. The major amendments to the text have been highlighted for ease of reference.

Thank you once again.

Reviewer 2 Report

In this paper, the authors investigated the effluent (secondary stage treated wastewater) post-treatment or removal of contaminants through the unsaturated and saturated zones in a large-scale pre-pilot experimental model. The topic is really interesting and relevant to this journal. But the specific comments are as follows:

  • The title of the paper is too long.!
  • The literature review should be improved by a brief analysis of new related articles.
  • The quality of Figure 1 should be improved.
  • Table 3 should be rewritten. Some parameters are not visible!!
  • Page 8, line 238: “Where:Wi - Weight of parameter in - Number of parameters γi -Index value for parameter i based on index curve” is not clear!!
  • Page 8, line 240: why did you add (1) to this sentence?
  • Page 8, line 244: “Figure 3-12” should be changed to “Figures 3-12”
  • Page 13, line 257: “Table 8 and 9” should be changed to “Tables 8 and 9”
  • The presentation of the paper should be improved.

Author Response

Dear Reviewer

First and foremost, we would like to take this opportunity to appreciate the efforts of the respected reviewers. We have attempted to follow the points carefully and modify the paper accordingly. Through a checklist as below we have included these modifications.

Best Regards

Mohsen Najarchi

----------------------------------------------------------------------------------------

For Reviewer No. 2:

The comments and cases mentioned by the second reviewer, which needed to be corrected in the text of the article, were made and the necessary edits were implemented in the text of the article. Due to its length, the title of the article was summarized as follows.

"Post-treatment of reclaimed municipal wastewater through unsaturated and saturated porous media in a large-scale experimental model"

We hope that the above statements have addressed the concerns brought up, and t----------------------he revised manuscript has improved for a positive recommendation by the Referees. The major amendments to the text have been highlighted for ease of reference.

Thank you once again.

Round 2

Reviewer 2 Report

The authors have addressed the comments. I suggest the manuscript be accepted for publication in Water Journal.